# Do Epstein–Barr Virus Mutations and Natural Genome Sequence Variations Contribute to Disease?

**DOI:** 10.3390/biom12010017

**Published:** 2021-12-23

**Authors:** Paul J. Farrell, Robert E. White

**Affiliations:** Department of Infectious Disease, Imperial College London, London W2 1PG, UK; robert.e.white@imperial.ac.uk

**Keywords:** Epstein–Barr virus, lymphoma, nasopharyngeal carcinoma, EBER, vaccine

## Abstract

Most of the world’s population is infected by the Epstein–Barr virus (EBV), but the incidence of the diseases associated with EBV infection differs greatly in different parts of the world. Many factors may determine those differences, but variation in the virus genome is likely to be a contributing factor for some of the diseases. Here, we describe the main forms of EBV genome sequence variation, and the mechanisms by which variations in the virus genome are likely to contribute to disease. EBV genome deletions or polymorphisms can also provide useful markers for monitoring disease. If some EBV strains prove to be more pathogenic than others, this suggests the possible value of immunising people against infection by those pathogenic strains.

## 1. Epstein–Barr Virus Biology 

Most people become infected by the Epstein–Barr virus (EBV, human herpesvirus 4) early in life, without any symptoms. Like all herpesviruses, EBV can infect cells at a peripheral location in the body to allow transmission, but also infect another cell type where latent (quiescent, non-productive, persistent) infection occurs. For EBV, transmission is via saliva transfer (for example, in kissing), with the virus perhaps briefly replicating in epithelial cells in the mouth or throat. This then gives access to the main target cells, which are B lymphocytes, most likely encountered in the tonsils or other lymphoid organs in the oropharynx. The infected B lymphocytes become activated and proliferate, but can then differentiate into memory B cells, in which EBV persists latently in vivo. The primary infection is normally controlled by the host’s immune response, which makes the natural killer (NK) cell and cytotoxic T cell responses that destroy the EBV-infected B cells, in addition to antibodies targeted to many viral proteins. In contrast to infants (who do not develop obvious symptoms), people who become infected as teenagers or adults may suffer infectious mononucleosis; their inflammatory symptoms correlate with an elevated CD8 T cell response against the virus [1]. Usually, those symptoms fade after a few weeks or months, and the recovered patient (or asymptomatically infected child) carries the virus for the rest of their life. People shed highly variable levels of infectious virus in their saliva over time [2] and maintain a long-term immune response, including antibodies to the EBNA1 viral protein and memory T cells that can target several EBV antigens.

Infection of B lymphocytes is accompanied by the expression of a set of viral genes that encode proteins (various EBNAs and LMPs), two functional small RNAs (EBERs) and many viral micro RNAs [3]. EBNAs are the Epstein–Barr nuclear antigens—EBNA1, EBNA2, EBNA3A, B, C and EBNA-LP; these proteins promote activation, proliferation and survival of the infected cell, and maintain the viral genome as a multicopy episome in the cell nucleus. In cell culture, EBV-infected primary B cells grow indefinitely as lymphoblastoid cell lines (LCLs). Thus, together, the viral EBNA and LMP proteins cause the cells to behave as if they had been activated by binding their cognate antigen.

Although most infected people will never experience any symptoms or disease from EBV infection, there are many diseases in which EBV plays an important role. The abilities of EBV gene products to cause cell proliferation, prevent apoptosis or evade immunity link the virus to some of these diseases. Details of EBV biology and its mechanisms in human cancer have been described elsewhere [3,4]. This article will focus on the significance of diseases caused by variations and alterations in the EBV genome.

### 1.1. Natural Variation in the EBV Genome

Understanding of EBV genome variation has progressed rapidly over the last 10 years since the development of the hybrid enrichment sequencing method [5]. This technique allows specific purification of the EBV DNA present in cells in culture or clinical samples for sequencing using standard methods [6]. There are now more than 1000 EBV genome sequences in Genbank, derived both from healthy people and from many of the diseases associated with EBV infection, with representation from many parts of the world.

In healthy carriers of the virus, and in most cases of diseases associated with EBV, the viral genome appears to be intact (without obvious deletions or rearrangements). There are, however, some examples where deletions occur that seem to be relevant to the disease: these can suggest mechanisms that cause disease. DNA viruses like EBV generally have quite stable genome sequences, but there is some variation between different parts of the world that might be relevant to the different geographic incidences of diseases associated with EBV. There is evidence for virus recombination [6,7] in EBV, but it is difficult to define evolutionary lineages of EBV because of the uneven distribution of polymorphisms and inter-strain recombination.

### 1.2. Type 1 and Type 2 EBV 

The classification of EBV into type 1 and type 2 strains, and some indication of geographic variation in the type 1 viral genome sequence, have been known since the 1980s. However, the greatly increased number of sequences in recent years has allowed the construction of a multiple sequence alignment of 241 EBV genome sequences [8], which has been used to define the basis of these classifications more clearly.

Comparison of single nucleotide polymorphisms (SNPs) between type 1 EBV and type 2 EBV strains has confirmed that the differences are mostly in the EBNA2 and EBNA3 genes. In the genome region around EBNA2, the differences are confined to the region between the last two coding exons of EBNA-LP (Y exons) and the protein coding sequence of EBNA2. The large internal repeat (IR1) immediately upstream, and the downstream repeat sequence encoding BHLF1 (called IR2 or the Not I repeat) cannot be used to distinguish type 1 from type 2 EBV [7]. At the EBNA3 locus, differences between type 1 and type 2 extend to the adjacent viral genes [9]. Interestingly, these encode the two glycoproteins on the surface of the virus particle (gp350 and gp42) that mediate binding to the EBV receptors found on B cells of CD21 and MHC class II, respectively.

EBNA2 functions as a transcription activator, switching on cell and viral genes required for cell growth [3]. One apparent counter-example to this, where EBNA2 was found to result in downregulation of the cell CIITA factor (which causes expression of MHC class II), was shown to be a consequence of EBNA2 activating the adjacent cell gene DEXI with the consequent reorganisation of cell chromatin indirectly resulting in the downregulation of CTIIA [10]. The EBNA3 genes also control cell gene expression, but have more complex mechanisms of action, which involve the repression or activation of many cell genes [11].

The best-known biological difference between type 1 and type 2 EBV is that type 1 strains are better at transforming B cells into LCLs than type 2 strains [12]. Replacing the EBNA2 of a type 2 strain with a type 1 EBNA2 is sufficient to convert the B cell transformation activity of the type 2 strain to that of a type 1 strain [13]. Although the amino acid sequences of type 1 and type 2 EBNA2 are only about 50% identical, the key determinant of the superior ability of type 1 EBNA2 to maintain LCL growth has been mapped to a few amino acids in the transactivation domain [14,15]. These amino acids control interaction with the cell protein BS69 (also called ZMYND11), which normally tends to inhibit the transactivation of function of EBNA2. There is an extra binding site for BS69 in type 2 EBNA2, which consequently appears to be inhibited more effectively by BS69 than type 1 EBNA2 [15]. EBNA2 function is also regulated by CDK1- and PLK1-dependent phosphorylation in the C terminal part of the protein, near the transactivation domain [16]. The EBNA2 amino acids that are phosphorylated by CDK1 and PLK1 are conserved in type 1 and type 2 EBNA2, but it is not currently known whether the differential binding of BS69 might modulate the EBNA2 phosphorylation.

Although homologous recombination appears to occur in most parts of the viral genome [6], detectable when cells have been infected by two or more strains of EBV, the EBNA2 and EBNA3 loci maintain their type 1 or type 2 identity in naturally occurring strains. This is thought to be because the EBNA2 and EBNA3 sequences are sufficiently different between the two types to prevent homologous recombination within these genes. Interestingly, intertypic recombinants (for example a virus with a type 2 EBNA2 and a type 1 EBNA3) are rare [8,17], suggesting that there may be also some functional compatibility required for cooperation between the EBNA2 and EBNA3 proteins of the same type that selects against such recombinants. This could be consistent with EBNA2 and EBNA3 proteins both associating with DNA through their interactions with RBPJ (also called CBF1). Chromatin immunoprecipitation (ChIP) analyses that have shown that they frequently bind to many of the same gene loci in the human genome [18,19].

Type 1 EBV is much more frequent than type 2 in most parts of the world, but in some parts of central Africa (Kenya, for example) there is an approximately equal prevalence of type 1 and type 2 EBV [17]. Some other tropical regions of the world are also known to have substantial levels of type 2 EBV, for example Papua New Guinea [9]. How the types arose and diversified is not known but, given its high presence in the tropics, it might be that the type 2 EBV found some historical selective advantage in the immune state of people who are chronically exposed to infection, for example, endemic malaria or other tropical diseases. Type 1 EBV strains are known to efficiently transform resting B cells but not exogenously activated B cells [20]; the activity of type 2 EBV in this comparison has not yet been investigated. Although the highest incidence of type 2 EBV is found in black African people, ethnicity is not a definitive determinant of type specific infection: some clear examples of white British people shedding type 2 EBV were found in a survey of university students in London [9].

Induction of the lytic replication cycle is initiated by activation of the expression of the EBV BZLF1 gene via its promoter Zp. The BZLF1 protein (also called Zta or ZEBRA in some publications) is a transcription factor. It switches on the early genes of the EBV lytic cycle, whose products mediate viral DNA replication, which then allows for the subsequent production of new virus particles. The most efficient and physiologically relevant mechanism for inducing the EBV lytic cycle from latency is via the signal transduction from the B cell receptor (BCR). This process can readily be activated in cell culture by cross-linking the BCR on the cell surface of some Burkitt lymphoma cell lines using a suitable isotype specific antibody, for example in the Akata Burkitt lymphoma cell line (which contains EBV). An EBV single nucleotide polymorphism (SNP) in Zp known as V3 results in the stronger activation of BZLF1 because it creates an extra binding site for the cell NF-AT transcription factor [21], which is activated by BCR signal transduction. This V3 polymorphism is present in all type 2 EBV strains, and about 50% of type 1 EBV strains, including Akata EBV [9]. LCLs containing type 2 EBV also have a higher expression of NF-AT, so this all results in type 2 EBV LCLs having more spontaneous lytic replication of EBV than type 1 EBV LCLs [22]. The shedding of EBV DNA in saliva was not found to be higher in people with type 1 V3 EBV SNP [9], but it is possible that the V3 SNP might increase EBV levels (or EBV lytic cycle gene expression) in vivo, in some locations in the body.

### 1.3. Geographic and Ethnic Variation in EBV Genome Sequence

Interest in EBV genome sequence variation developed from the need to ensure that the lab strains used for studying the virus are representative of clinical isolates and from large geographic variations in the incidence of some diseases associated with EBV. For example, nasopharyngeal carcinoma (NPC) has a high incidence in South China and EBV-associated Burkitt lymphoma is frequent in sub-Saharan Africa (reviewed in [3]). Among the many possible reasons for those different disease incidences (for example, environmental carcinogen exposure and host genetics), geographic variation in the virus genome would be an important consideration.

Principal component analysis (PCA) has proven to be an effective method for analysing EBV genome sequence variation. Based on a multiple sequence alignment (MSA) of 241 EBV genomes from samples that were annotated with geographic region of origin or ethnicity of the infected person, PCA clustering showed that there are systematic geographic differences [8]. The results showed that the EBV sequences clustered into distinct geographic categories, for example, Asia, Europe, USA, Africa, South America and Papua New Guinea. Similar clustering patterns were seen using phylogenetic tree analysis of the sequences [8], but the phylogenetic trees did not provide a method to identify which nucleotides in the EBV genome determined the clustering.

As the values for each sequence on each principal component (eigenvector) in the usual PCA clustering plots are derived by summing the individual PCA coefficients at each genome position where the nucleotide in that sequence differs from the consensus, the contribution of each nucleotide position to each component can be quantified. Most of the coefficients are very small but a few locations in the sequence have relatively large coefficients, and these can be used to identify combinations of nucleotide positions where SNPs are characteristic of each geographic cluster [23]. Thus, the EBV genome sequence at some of the nucleotide positions characteristic of each geographic cluster (Figure 1) allows assignment of a new EBV sequence to a geographic category. This may prove particularly useful for samples that may contain too little EBV to determine the complete genome sequence, but sufficient to PCR amplify and sequence in some small regions containing these diagnostic SNPs.

EBV persists in an infected person in a balance with host immune surveillance, so it seems likely that positions of positive selection in the virus genome sequence might correspond to epitopes for cytotoxic T cell surveillance. EBNA3 proteins are major targets for this immune surveillance, and CD8 T cell epitopes have been mapped extensively in this gene region in patients from different parts of the world, and with a range of known HLA types. Surprisingly, there was little correlation between the points of positive selection in the EBNA3 genes and the characterised immune epitopes [6]. However, high levels of linkage disequilibrium appear to be slightly more common in EBV genes that contain known T cell epitopes, which suggests that host immune selection has probably contributed to patterns of genetic diversity in EBV [24].

In summary, we know that variation at defined nucleotide positions in the EBV genome is characteristic of the geographic region of origin of EBV strains, but there is uncertainty about the factors that select those differences. It is useful to note that direct sequencing of the EBV DNA from normal, healthy, persistently infected people shows that they usually contain a predominant unique EBV sequence, but there can also be low levels of alternative sequences, indicating mixtures of strains in some individuals. It is not known whether those mixtures derive from being initially infected with a mixture of strains, or are due to superinfection during normal social contact. This question of whether normal immunity protects against additional infection will become an important point in the context of potential future vaccines against EBV.

## 2. Sequence Variations in Diseases Associated with EBV

### 2.1. Immune Deficiency in EBV-Associated Diseases

Immune suppression resulting from inherited immunodeficiency genes, HIV infection or immunosuppressive therapy after a tissue transplant may allow the B blasts that result from EBV infection of normal B cells to cause a lymphoproliferative disease. In this case, the viral gene expression is similar to that of an LCL in culture, with all the EBNAs, LMPs and non-coding RNAs. Healthy carriers of EBV have abundant cytotoxic T cells specific for epitopes in the EBNA and LMP proteins. The excessive proliferation of EBV infected lymphoblasts in transplant patients (post-transplant lymphoproliferative disease—PTLD) is a significant clinical problem. Partial reduction of the immunosuppressive treatment may allow the patient to recover, eliminating the excessive EBV-positive B blasts without rejection of the transplant. Some centres can also grow EBV-specific cytotoxic T cells, either from the patient’s own blood or from a panel of HLA-matched donors, and use these for successful therapy [25].

The unrestrained proliferation of EBV-positive Blasts is also a feature of X-linked lymphoproliferative syndrome (XLP), which is due to an inherited mutation in the SH2D1A gene. SH2D1A encodes SAP, a component of lymphocyte signaling complexes in T cells and NK cells that is crucial for their development. Boys who inherit the mutation can develop a potentially fatal lymphoma due to defective immune surveillance and the proliferation of EBV-infected B lymphocytes [26]. Many of the relevant families are now identified and the affected children may receive a haematopoietic stem-cell transplant to prevent the disease. There is currently no evidence for any role of specific EBV variants in XLP patients.

In some other immune deficient patients, the genetic defect is not in the production of the cytotoxic T cells that would normally kill EBV-infected cells, but in their function [27]. In this situation, an ineffective immune response to EBV-infected cells causes excessive recruitment of macrophages by gamma interferon from the T cells that can result in a severe inflammatory disease called haemophagocytic lymphohistiocytosis (HLH).

Before modern combinations of anti-HIV drugs were developed, oral hairy leukoplakia was a frequent sign of advanced AIDS disease. This appeared as white patches, usually on the edge of the tongue or some other oral mucosal sites. Biopsies showed that EBV was replicating in defined layers of the stratified epithelium [28], presumably made macroscopically visible by the lack of immune surveillance in AIDS immunodeficiency. This is the clearest opportunity to observe epithelial replication of EBV in vivo. The sites where EBV replication occurs in healthy infected people producing the usual low-level or sporadic shedding of virus into saliva cannot normally be recognised [2], so these oral hairy leukoplakia lesions may give some insight into that process.

#### EBV Genome Deletions in T/NK Disease 

Although the main life cycle of EBV occurs in B lymphocytes and most EBV associated lymphomas are in B cells, a low incidence of lymphomas with EBV infection of T cells or NK cells also occurs. The biology of EBV in T cells remains uncertain: some strains of type 2 EBV have been shown to infect adult T cells in vitro [29] and EBV is also readily detected in the T cells of Kenyan infants that have been infected with type 2 EBV [30]. However, most T/NK cell malignancies contain type 1 EBV. These lymphomas often arise in children, who are sometimes found to have a mutation in a gene involved in immune regulation [27]. Although T cells and NK cells from adults do not normally express the usual B cell receptor for EBV infection (CD21), these NK/T lymphomas show that infection of the cells must have occurred somehow. Immature T cells in the adult thyroid express CD21, and in very young children most circulating T cells express CD21 [31], so it is possible that this accounts for the fact [32] that T cells can be infected by certain strains of EBV in culture (via CD21). The direct infection of NK cells in culture has not been possible, but NK cells have been found to use antibodies to capture and internalise membrane-bound EBV particles (with CD21) from the surface of B lymphocytes, so this might offer an additional mechanism [33]. Infection of the common progenitor of T and NK cells in haematopoietic development has also been proposed as a mechanism for the rare viral entry that occurs into these cell types that might precede NK/T cell lymphoma development [34]. 

EBV T/NK lymphomas and chronic active EBV infection (CAEBV) involving the infection of T cells appear to be more frequent in Asia (Japan, China, Korea) than Western countries, but many cases have now also been described in Europe [35] and the USA [36]. In a about 30–40% of T/NK lymphomas, there is a deletion or re-arrangement of the EBV genome [35,37,38]. Often, a wild-type EBV genome is detectable in addition to the deleted genome. These mutations in the EBV genome are different in every patient, but they are most frequently clustered around the BART transcript region, and seem likely to affect the expression of the BART miRNAs. How these mutations promote disease is not fully understood, but experiments comparing wild-type EBV with strains lacking the BART region in infection of peripheral blood mononuclear cells (PBMCs) from adolescents indicated that absence of the BART miRNAs resulted in an interferon response [39]. The patients who develop these diseases are often found to have inherited immune deficiencies affecting T or NK function [27], so the mechanism likely involves a combination of immunodeficiency and the altered EBV gene expression in a cell type that is not usually infected by EBV. 

### 2.2. Lymphomas Derived from B lymphocytes

#### 2.2.1. Burkitt Lymphoma

EBV was originally discovered in the early 1960s using electron microscopy to search for viral particles in Burkitt lymphoma. The incidence of Burkitt lymphoma (BL), particularly in children, is high in central Africa and some other parts of the world where malaria is endemic. The high incidence of BL in these areas led to Denis Burkitt identifying the “endemic BL”: in these cases, EBV is almost always present in the malignant cells. In other parts of the world, “sporadic BL” tumours are rare, and usually do not contain EBV (although the patients are typically infected by EBV). However, the phenotypic characteristics of BL are primarily defined by their EBV status, regardless of whether they originated in an endemic or sporadic area [40].

BL cells are now known to always have a chromosome translocation which places the *MYC* proto-oncogene under the transcriptional control of one of the immunoglobulin gene loci, resulting in an abnormally high production of MYC protein. The high level of MYC activates cell proliferation, but the associated proliferative stress also induces apoptosis. EBV-negative tumours usually have a mutation in the p53 pathway (TP53, USP7 or CDKN2A mutation in 75% of cases) that overcomes this tendency to undergo apoptosis, but these occur in only 30% of EBV-positive BL [40]. This is because EBV has evolved a variety of mechanisms to overcome the apoptosis, discussed in detail in [3], that can help the BL cells to survive. In some BL tumours, there is a deletion in the EBV genome which removes the EBNA2 region, leading to altered splicing of BHRF1 transcripts under the control of the Wp latent cycle EBNA promotor, and the increased production of BHRF1 protein [41]. BHRF1 is a paralogue of the cell BCL2 protein, and has a powerful anti-apoptotic function. Examples of these EBV genome deletions were known for many years in the P3HR1 and Daudi BL cell lines (Figure 2), but they have now been found to be present in about 10% of BL cell lines analysed [41]. The altered splicing can also result in the expression of EBNA3 proteins in these BLs. Interestingly, type 1 EBV is found more often than type 2 in African BL (in Kenya), even though both virus types are present at about equal frequency in the population [17].

#### 2.2.2. Hodgkin Lymphoma 

In Western countries, the most frequent EBV-associated lymphoma is Hodgkin disease, in which about 30% of cases have EBV in the tumour-defining Reed–Sternberg cells. These are the key malignant cells in the cancer (much of the rest of the tumour is composed of reactive normal B cells). The Reed–Sternberg cell is thought to be derived from a B cell which has undergone a defective B cell receptor mutation or rearrangement that would be expected to result in cell apoptosis. In most cases of the EBV negative disease, these cells survive because of a mutation in the A20 apoptosis regulator, which affects signaling to NF-kB [42]. In the EBV-positive Reed–Sternberg cells, the viral LMP2A protein can produce survival signals including those provided by the BCR [43], whereas LMP1 is thought to provide signals that substitute for T cell help, preventing cell death and allowing the disease to develop. 

#### 2.2.3. Diffuse Large B Cell Lymphoma (DLBCL) 

The DLBCL classification is based on a cellular phenotype. However, EBV-associated DLBCLs are often defined by a contributory immunological defect, for example HIV lymphomas and post-transplant lymphomas (PTLD). Overall, EBV-associated DLBCL is a relatively uncommon category—around 3–5% of the total. There are two main phenotypic subtypes: activated B cell (ABC)-DLBCL and germinal centre DLBCL, based on the cell surface markers of the cell type. Based on the observation that a recombinant EBV, deleted for its EBNA-3B coding region, induced an ABC-DLBCL-like tumour in mice reconstituted with human immune cells, examples were identified of nonsense mutations in the EBNA3B gene in three ABC-DLBCLs, from PTLD or HIV lymphoma patients [44]. The extensive polymorphisms in the EBNA3B gene currently preclude an accurate assessment of the proportion of DLBCLs that carry an EBNA-3B mutation, but the frequency of potential EBNA3B mutations was substantially higher in ABC-DLBCL than other EBV cancers. Intriguingly, examples of small in-frame EBNA3B deletions have also been identified in Hodgkin and Burkitt lymphomas, suggesting a wider role for EBNA3B mutation in the induction of EBV-associated cancers [44]. More examples of BART region deletions (mentioned above in the context of NK/T lymphomas) have been found in DLBCL cases from Japan [38]. 

### 2.3. Carcinomas Associated with EBV 

The malignant cells of most cases of undifferentiated nasopharyngeal carcinoma and about 8% of cases of gastric carcinoma worldwide contain EBV and have some viral gene expression. Worldwide, these are the most frequent cancers associated with EBV and most cases occur in immunocompetent people. The diseases reflect the ability of EBV to infect epithelial cells as part of its normal life cycle. 

#### 2.3.1. Nasopharyngeal Carcinoma (NPC) 

NPC is exceptionally frequent in Southern China, having an incidence there about 50-fold higher than in Europe, but there are also regions of intermediate incidence in several other parts of the world. The tumour includes malignant epithelial cells containing EBV and a substantial lymphocytic component which is thought to facilitate tumour growth and immune evasion. The lymphoid component of NPC does not usually contain EBV. Polymerase chain-reaction (PCR) detection of EBV DNA in blood plasma is an effective way to identify patients with early-stage NPC in high incidence areas [45], when the disease can be treated successfully. 

There is no evidence for large scale deletions or rearrangements of the EBV genome in NPC, but the unusual geographic distribution of the disease has led to many investigations of SNPs or other small differences in the virus that might contribute to the disease. These studies have demonstrated the complexity of distinguishing a geographic virus sequence variation that is not relevant to disease from sequence changes that might give a more pathogenic virus. Several additional factors have been associated with NPC incidence, including the MHC variants prevalent in Cantonese people in South China who are at risk [46] and dietary carcinogens (although the traditional association of dried salted fish and other preserved foods with NPC has been questioned in a recent large study [47]). It has been difficult to distinguish whether the disease incidence is determined by these other factors, or whether there might also be a specific variant of EBV prevalent in the NPC endemic areas. 

Some of the EBV polymorphisms that have been proposed to be linked to NPC may be correlated to geographic or ethnic differences in EBV strains, rather than an effect of the disease. The Zp V3 polymorphism (which could be indirectly pathogenic by increasing the level of virus or EBV lytic cycle gene expression in an infected person) was epidemiologically linked to NPC cases in Indonesia [8]. The CAO deletion within the C terminal part of LMP1 is prevalent in China and Taiwan [23] and CAO LMP1 was found to be more oncogenic than the reference LMP1 in an epithelial cell culture assay [48], but has not been shown to be enriched in NPC cases relative to matched controls. LMP1 expression is quite variable in NPC tumours: it is now understood that NPC tumours require NF-kB expression, but this is achieved either through mutations in the NF-kB signaling pathway that result in NF-kB activation, or by LMP1 expression. One or the other of these two mechanisms is found in NPC biopsies [49]. 

A study of EBV sequence variation comparing geographically matched NPC cases and controls from South China (high-incidence region) found a SNP in the EBV RPMS1 open reading frame to be a specific risk factor for NPC [50]. This caused reduced stability of the RPMS1 protein, but there is unfortunately no evidence that the RPMS1 protein is produced in any EBV-infected cells [51]. Further sequencing studies found a SNP in the EBV BALF2 protein to be strongly associated with NPC cases [52], but no specific mechanism could be proposed. Since BALF2 is a single-stranded DNA-binding protein that functions as a processivity factor for the EBV genome replication complex, its expression could potentially facilitate oncogenesis by promoting aberrant recombination or reduced replication fidelity, but functional analysis of the variants will be required to understand whether this SNP might promote NPC development.

In contrast, a study in Hong Kong comparing EBV strains in NPC cases with healthy Cantonese people linked a polymorphism in the viral non-coding RNA EBER2 to NPC disease [53]. Recent molecular analysis of this M81 EBER2 polymorphism, which is frequent in Chinese EBV compared with other parts of the world, suggests a mechanism by which those viruses could be more oncogenic [54]. EBER2 is now known to form a ribonucleoprotein complex in the cell nucleus that affects gene expression. This was first described to affect the transcription of the EBV LMP2 gene by binding to the EBV terminal repeats [55], but further analysis has shown that it also increases expression of the cell UCHL1 gene [54]. Base pairing between part of the EBER2 RNA and the 5′ end of the nascent UCHL1 mRNA facilitates expression of the UCHL1 mRNA. UCHL1 is a deubiquitinase that increases the expression of cyclin B1 and Aurora B; both are cell cycle control genes. EBER RNAs with the M81 variant of EBER2 (which has a sequence change that increases its binding efficiency to the UCHL1 mRNA) resulted in about five-fold more Aurora B protein expression than the reference EBER genes [54]. This is the most convincing potential oncogenic mechanism related to EBV SNPs that has been found so far and may prove to be significant in the incidence of NPC. 

The large number of EBV strains already sequenced from China allows an assessment of the relative frequency of the M81 Chinese EBER2 variant and the Zp V3 SNP in NPC cases relative to normal controls (Figure 3). Of the 1015 EBV genome sequences in Genbank that could be classified according to their EBER and Zp SNPs, 500 were from China, of which 262 were NPC cases and 186 were healthy controls (Figure 3). The incidence of the reference Western strain B95-8 EBER2 and Zp was compared to incidence of the M81 EBER2 and Zp V3 SNP. The M81 EBER2 variant was found in 93% of Chinese NPC cases, but only 43% of isolates from healthy Chinese people. The Zp V3 SNP was enriched in NPC cases to a lesser degree; 78% of NPC cases and 52% of controls. Most strikingly, the combination of M81 EBER and V3 Zp was 22.5-fold more frequent in NPC cases relative to healthy controls than the B95-8 EBER and Zp SNPs (Figure 3). The data therefore support a connection between these polymorphisms and the incidence of NPC, particularly the EBER M81 variant. 

#### 2.3.2. Gastric Carcinoma

About 8% of gastric carcinoma cases have EBV in the malignant cells. A major study of genetic mutations in gastric carcinoma [56] showed that the EBV-positive cases are genetically and transcriptionally distinct, having high levels of promoter methylation, and frequent mutations in PI3kinase (activating the kinase) and ARID3A, but lacking the p53 mutations which are frequent in the EBV negative disease. At present, there is no evidence for EBV mutations in gastric carcinoma, and there are no reports of EBV SNPs specifically related to the disease. The incidence of gastric cancer is relatively high in Japan; the M81 EBER2 has a very low frequency there, but about half the reported sequences in Genbank have the Zp V3 polymorphism.

## 3. Multiple Sclerosis 

Of the many autoimmune diseases epidemiologically linked to EBV, multiple sclerosis (MS) is the most strongly associated. MS involves damage to the myelin sheath that protects neurons and consequently results in progressive loss of nerve function. The speed of progression and disease severity varies—often remission and relapses occur, but eventually the disease progresses in most cases. EBV has been suggested as a source of antigens that might result in an auto-reactive immune response, for example by cross reaction with proteins in cells important for nerve function. Studies of large numbers of cases and controls indicate that essentially 100% of people with MS have an EBV infection, compared with about 90% of the control population [57] and the cerebrospinal fluid (CSF) of MS patients is often found to contain oligoclonal immunoglobulins, consistent with immune cross-reaction contributing to disease. Surprisingly, EBV sequence variation has not been studied in relation to MS. Some EBV-infected cells have been found in the brains of MS patients [58,59,60], but the proposed role of EBV in this disease and mechanism remain to be established. One interpretation would be that EBV infection can be considered as essential for activating the host’s risk alleles [61], but a cross-reactive immune response to an EBV antigen that also recognizes a cell protein in the brain is very likely to mediate the disease in some cases. Recent studies, using an animal model system for MS, have proposed a new class of CD8+ suppressor T cells, which could play a role in controlling the disease [58]. Moreover, the administration of CD8 T cells has been investigated in MS patients, with some clinical responses noted [62]. 

## 4. Vaccines to Prevent Diseases Associated with EBV

Virus-associated cancers offer the possibility of preventing cancer by immunisation against virus infection, or by developing drugs targeting the virus. Antiviral treatments for hepatitis C and vaccines against Hepatitis B and human papillomaviruses can already prevent cancers associated with those infections, which are the leading viral causes of cancer. Vaccines or antivirals targeting EBV—the next biggest cause of viral cancers—have therefore become a high priority. 

In the past, most attempts at immunisation against EBV used immunogenic parts of the gp350 EBV surface glycoprotein, because antibodies to gp350 can neutralise infection in cell culture. An important early clinical trial of a gp350 vaccine in healthy EBV-negative people involved immunising half the group, while the others received a placebo. The health of the participants was monitored over 19 months, observing the outcome as the people carried on their normal lives and social contact [63]. The frequency of infectious mononucleosis was reduced in the immunised group, but there was no reduction in asymptomatic EBV infection. It is now recognised that antibodies to gH/gL and gp42 can also neutralise infection [64]. New clinical trials are therefore being developed using improved formulations of gp350, alongside some of these additional glycoproteins [65]. These vaccines seem likely to be able to prophylactically reduce the incidence of infectious mononucleosis in healthy teenagers, and to reduce post-transplant lymphoproliferative disease in solid organ transplant patients who were not already infected by EBV. However, if EBV infection of the person is not prevented by this type of immunisation, EBV-associated cancers occurring in immunocompetent people may not be reduced. Those cancers develop over many years, and EBV would likely still be present in the body. 

If endemic strains in some parts of the world are found to be more oncogenic (as discussed above for NPC), this could suggest a strategy of deliberately infecting infants with an attenuated non-oncogenic strain of EBV. This type of immunisation was done successfully for many years in some countries for another herpesvirus, Varicella Zoster virus (VZV), which causes chickenpox and shingles. That immunisation infects the child with the Oka strain of VZV, which causes a very mild infection that does not result in chickenpox symptoms, but can establish latency and give long term immunity. It would be a big step to consider doing the same with natural EBV strains, but the possibility of creating an attenuated version of one of these, or a virus-like particle, for immunisation is currently under investigation. Understanding worldwide EBV sequence variation might give a background of knowledge for developing a suitable virus vaccine strain, but it is also not currently known whether being infected with EBV protects against superinfection. If it does not, then the endemic (oncogenic) strain would likely still become established, and there would probably be little advantage from the immunisation. It will therefore be important to establish whether EBV infection confers resistance to superinfection. 

## 5. Conclusions 

The improved sequencing of EBV DNA directly from clinical samples using the hybrid enrichment method has provided a much greater understanding of the variation in the virus genome and the deletions and polymorphisms that are associated with some EBV diseases. This helps to focus research on virus genes that are important for disease and the mechanisms of these diseases. In principle, PCR to detect EBV genome deletions can be used for monitoring the disease load (for example, in some NKT lymphomas), but the different deletions in every case reduce the convenience of this approach. 

For a link between sequence variation and disease to be convincing, both an epidemiological association and a mechanism are required. Differences in the activities of EBNA2 and EBNA3 proteins may explain the superior transformation of B lymphocytes by type 1 EBV strains, but this has not yet been linked to an EBV-associated disease. The recently identified functional differences in EBER2 function between Chinese and African/European EBV strains (with M81 as the prototype Chinese example) appear epidemiologically to contribute to the elevated incidence of NPC in Southern China. To fulfil Koch’s postulates, this will need to be demonstrated in an experimental cancer induction model. If the pathogenic nature of some EBV strains is confirmed, this will strengthen the case for immunisation against infection by those strains. 

## Figures and Tables

**Figure 1 biomolecules-12-00017-f001:**
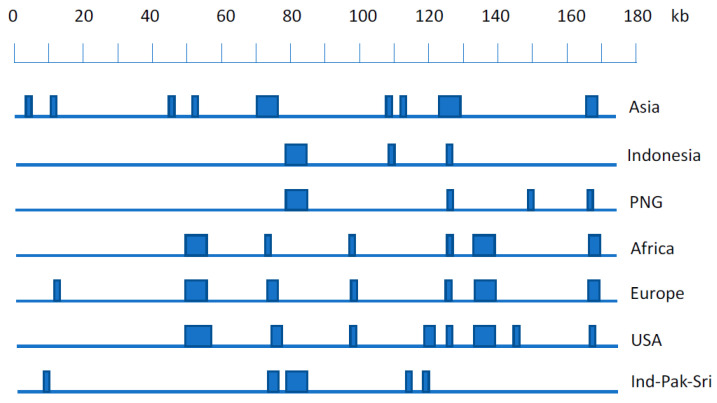
Schematic showing EBV genome locations containing SNPs characteristics for geographic origins of EBV on a scale of the reference NC007605 EBV genome. Vertical bars represent genomic positions of SNPs or haplotypes that are geographic location-specific. Exact details of SNPs specific for each geographic location are given in Table S3 of [23], (details of Ind-Pak-Sri locations will be published elsewhere, but are available on request). PNG is Papua New Guinea; Ind-Pak-Sri is India, Pakistan, Sri Lanka.

**Figure 2 biomolecules-12-00017-f002:**
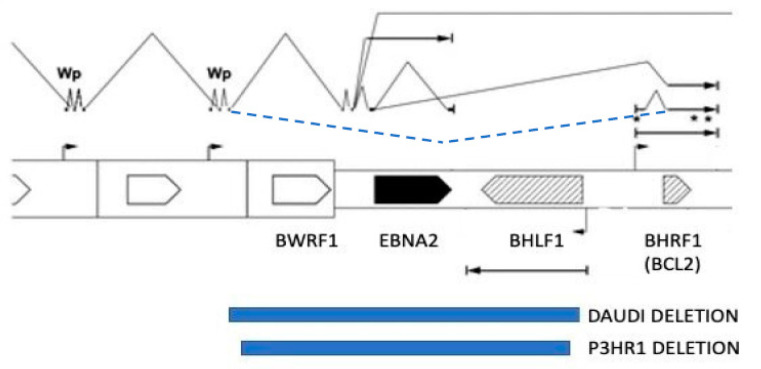
Genomic locations of Daudi and P3HR1 BL deletions (blue boxes) that result in increased expression of the BHRF1 protein from the spliced mRNAs, which normally express the EBNA proteins. Black lines above the open reading frame map indicate the spliced transcripts produced by intact EBV genomes. The novel RNA splicing from the W2 exon near Wp to the BHRF1 coding exon in BL cells with this deletion [41] is shown as a dashed line (blue). Symbols * are BHRF1 miRNAs.

**Figure 3 biomolecules-12-00017-f003:**
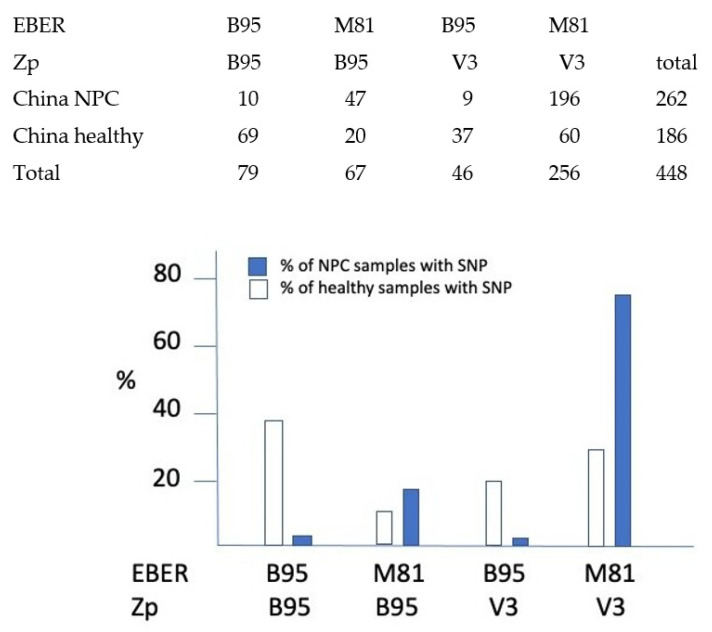
Incidence of the different combinations of EBER2 and Zp SNPs in Chinese EBV strains, comparing viruses from NPC disease with healthy controls. *x*-axis shows the different possible combinations of Zp and EBER SNPs, classified into either B95 (the sequence of the reference B95-8 western strain) or the Zp-V3 and EBER2-M81 SNPs described in the text. Chi squared test shows a highly significant increase (*p* = 1 × 10^−27^) in NPC cases in people whose EBV contains both EBER2-M81 and Zp-V3 SNPs.

## Data Availability

Not applicable.

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
