# Peer review of "Do Epstein–Barr Virus Mutations and Natural Genome Sequence Variations Contribute to Disease?"

_biomolecules, 2021, doi:10.3390/biom12010017_

Round 1
Reviewer 1 Report
This review is focuses on the role that mutations or naturally occurring variations in the Epstein-Barr virus genome play in the diseases associated with EBV infection. It provides a comprehensive overview that is inclusive, easy to read and very well written, and it will be very useful to investigators interested in EBV.
Minor points:
- Denis Burkitt is listed as Dennis Burkitt (line 298).
- In Wp BL, the EBNA3 proteins are expressed in addition to BHRF1. There is some data that suggests that these proteins may also play a role in the growth of the tumor cells.
Author Response
Minor points:
1. Denis Burkitt is listed as Dennis Burkitt (line 298).
Response – corrected. Thank you for pointing this out.
2. In Wp BL, the EBNA3 proteins are expressed in addition to BHRF1. There is some data that suggests that these proteins may also play a role in the growth of the tumor cells.
Response – added “The altered splicing can also result in expression of EBNA3 proteins in these BLs.”
Reviewer 2 Report
In this review, the authors describe the main forms of EBV genome sequence variation and mechanisms by which variations in the virus genome are likely to contribute to different diseases.
The review is well organized and comprehensive.
Author Response
No specific revisions required
Reviewer 3 Report
The manuscript provides an interesting review of the status of research on EBV genome variations and the relationship between the geographic distribution of specific polymorphisms and EBV-associated diseases. The paper is well documented and clearly written.
Mino points:
- Lane 108-109 – the authors mention that homologous recombination is relatively frequent in most parts of the EBV genome except the EBNA coding regions. A short mention of the possible mechanism of homologous recombination and its possible relationship with co-infection with different EBV stains would be useful.
- Lane 227 – the authors mention that EBV immortalized LCLs express latency III. This may be quite confusing for the reader since the concept of different types of latency was not introduced earlier in the manuscript.
- Lane 280-282 – the authors mention that deletion or rearrangement are detected in a significant proportion of T/NK lymphomas. A more precise statement on the frequency of these genetic abnormalities would be preferable.
- The authors include a very brief chapter on the relationship between EBV infection and multiple sclerosis. This is a very interesting topic that is receiving much attention, but the short chapter gives only a very superficial description of the relevant issues, none of which is related to genome sequence variations. The authors should consider omitting this section.
Author Response
Minor points:
1. Line 108-109 – the authors mention that homologous recombination is relatively frequent in most parts of the EBV genome except the EBNA coding regions. A short mention of the possible mechanism of homologous recombination and its possible relationship with co-infection with different EBV stains would be useful.
Response - Line 108 sentence modified to:
Although homologous recombination appears to occur in most parts of the viral genome [6], detectable when cells have been infected by two or more strains of EBV, the EBNA2 and EBNA3 loci maintain their type 1 or type 2 identity in naturally occurring strains
2. Line 227 – the authors mention that EBV immortalized LCLs express latency III. This may be quite confusing for the reader since the concept of different types of latency was not introduced earlier in the manuscript.
Response - Yes, Latency III is unnecessary; removed.
3. Line 280-282 – the authors mention that deletion or rearrangement are detected in a significant proportion of T/NK lymphomas. A more precise statement on the frequency of these genetic abnormalities would be preferable.
Response – changed to: In a about 30-40% of T/NK lymphomas, there is a deletion or re-arrangement of the EBV genome [35, 37, 38].
4. The authors include a very brief chapter on the relationship between EBV infection and multiple sclerosis. This is a very interesting topic that is receiving much attention, but the short chapter gives only a very superficial description of the relevant issues, none of which is related to genome sequence variations. The authors should consider omitting this section.
Response – the important and relevant point that we want to make is in the sentence: “Surprisingly, EBV sequence variation has not been studied in relation to MS.”
I am aware of a major new publication coming in early 2022 which will greatly increase interest in EBV and MS, so I want to ensure this review is topical by retaining this short section on MS.